# Social support and very young adolescent girl's knowledge on sexual relationships: A comparative qualitative study of Girl Only Clubs' participants and non-participants in rural Malawi

Wanangwa Chimwaza-Manda [1,2] *, Mphatso Kamndaya [3], Nanlesta Pilgrim[4¤], Sanyukta Mathur[4], Effie Kondwani Chipeta[2], Yandisa Sikweyiya [1,5]

1 School of Public Health, University of the Witwatersrand, Johannesburg, South Africa, 2 Kamuzu University of Health Sciences, Blantyre, Malawi, 3 School of Applied Sciences, University of Malawi-The Polytechnic, Blantyre, Malawi, 4 Population Council, Washington, DC, United States of America, 5 Gender and Health Research Unit, South African Medical Research Council, Pretoria, South Africa

¤ Current address: ViiV Healthcare, London, England, United Kingdom
* wmanda@cartafrica.org

## Abstract

Early sexual relationships are associated with an increased risk of acquiring sexually transmitted diseases including HIV/AIDs, teenage pregnancies, and unsafe abortions among other negative health outcomes. Understanding sexual relationships among very young adolescents (VYAs) is important to equip them to protect themselves from negative sexual health (SH) outcomes. DREAMS (**D**etermined, **R**esilient, **E**mpowered, **A**IDS-free, **M**entored, and **S**afe) is an HIV prevention initiative that provided an evidence-based core package of interventions to VYAs to prevent HIV acquisition in 15 countries in sub-Saharan Africa. The Girl Only Club (GOC) was the primary context for the interventions. Our objective in this study was to explore if there was any difference in social support (SS) received concerning sexual relationships between the VYA girls who attended GOCs and those who did not. In-depth interviews were conducted with 43 VYA girls, aged 10–14 years, in two rural southern districts, Zomba and Machinga, in Malawi. Twenty-three VYA girls were participants in GOCs and 20 VYA girls did not participate. A thematic, descriptive approach that involved a constant comparative analysis guided the data analysis, and Nvivo 12 software was used. In both study sites available SS concerning sexual relationships is informational support including information from parents, older relatives, and friends. However, club participants differed from non-club participants in sexual and reproductive health (SRH) knowledge and use. Club participants reported consulting others on decision-making and information on sexual relationships; receiving detailed SH information from clubs; condom use due to education received from the clubs; quitting sexual relationships; and correcting misinformation with club information. GOC participants received more SS which made them more knowledgeable and better at handling sexual relationship issues than those not in

**Data Availability Statement:** All relevant data are within the paper and its Supporting information files.

**Funding:** Funding for the study was provided in part by the generous support of the American people through the United States President's Emergency Plan for AIDS Relief (PEPFAR) and the United States Agency for International Development (USAID) under Project SOAR (Cooperative Agreement AID–OAA–A–14–00060). The contents of this manuscript are the sole responsibility of the authors and do not necessarily reflect the views of PEPFAR, USAID, or the United States Government. In addition, this research was supported by the Consortium for Advanced Research Training in Africa (CARTA). CARTA is jointly led by the African Population and Health Research Center and the University of the Witwatersrand and funded by the Carnegie Corporation of New York (Grant No. G-19-57145), Sida (Grant No:54100113), Uppsala Monitoring Center, Norwegian Agency for Development Cooperation (Norad), and by the Wellcome Trust [reference no. 107768/Z/15/Z] and the UK Foreign, Commonwealth & Development Office, with support from the Developing Excellence in Leadership, Training and Science in Africa (DELTAS Africa) Programme. The statements made and views expressed are solely the responsibility of the author.

**Competing interests:** The authors have declared that no competing interests exist.

clubs. Interventions that integrate SS including social asset building and safe spaces are critical for VYA SRH programming.

## Introduction

Early adolescence, the period between the ages of 10 and 14, is characterized by enormous biological, cognitive, sexual, emotional, and social changes [1, 2]. These include the development of secondary sexual characteristics such as breast budding, rounded figure or hips and experience of menstruation in girls, ejaculation in boys and most attain sexual maturity [1, 3, 4]. During this period, adolescents also begin to form new gender roles and sexual identities within their social contexts [5]. As these developments occur, very young adolescents (VYAs) also acquire information, develop attitudes and experiment with behaviors and sexual relationships [1, 6].

Although the majority of VYAs do not engage in penetrative sex, their sexual curiosities have begun and some have reported engaging in non-penetrative sexual activities such as kissing, fondling, foreplay, and heavy petting [7–10]. The few studies that have been conducted among VYAs in sub-Saharan Africa (SSA) reveal that substantial numbers of boys and girls in many countries engage in unprotected heterosexual vaginal intercourse either by choice or coercion––before their 15th birthday [11–14]. These behaviors predispose VYAs to negative SRH outcomes such as unwanted pregnancies, HIV, and other sexually transmitted infections [1, 2].

Globally studies conducted on adolescent SH have shown that various forms of SS within the social contexts in which adolescents grow and develop play an important role in their SH practices and behaviours [15–20]. However, there are a few studies that have explored how the VYAs experience use of these various forms of SS to address their SH needs. In addition, there remains scant information in SSA about how VYAs who are engaged in sexual relationships use SS systems to navigate through their relationships including their decision-making about sexual intercourse, and how they protect themselves from negative SRH outcomes, such as unwanted pregnancy or sexually transmitted infections [21]. SS is a multifaceted concept that has been widely studied over the years. It has however been defined and measured and differently by various researchers [22–24]. Despite various definitions, some scholars concur that SS includes some form of positive interaction or offer of help to someone in need of support [24–27]. For instance, in one study they used a definition by Lin et al. and defined SS as the instrumental and/or expressive provisions, real or perceived, given by the community, social networks, and intimate relationships [23, 28]. In another paper, Braga et al. used a definition by another author and referred to SS as relationships among people, to the structure and quality of these relationships, and concrete actions such as material help, sharing of information, being supportive in critical moments of life and sensitive to the perceptions of people concerning all these aspects, which enables them to have greater autonomy and, consequently, satisfactorily respond to intrusive experiences [24]. In this study, we used Barker's definition in which SS is understood as a range of interpersonal relationships or connections that have an impact on the individual's functioning, and generally includes support provided by individuals and by social institutions [29]. The source of this SS includes parents, peers, extended family, and social institutions such as youth clubs among others.

Considering their age, VYAs are less likely to access SRH information and services from formal sources [2]. In addition, most adolescent SH interventions focus on older adolescents

overlooking the SH needs of VYA [2, 8, 9, 30]. For instance, in Malawi few adolescent health programs reach VYA with SRH information, life and relationship skills, and violence prevention education, which are key elements considered to reduce HIV risk (e.g., delayed sexual debut) and empower girls [31]. Furthermore, little is known about VYA's experiences with such programs and their impact on their lives [31]. Learning more about such VYAs programs has the potential to contribute to the evidence that is required to develop evidence-based, culturally, and age-appropriate SH programs for addressing the SH needs of VYA in Malawi.

An example of an SH intervention for VYAs implemented in Malawi is the Go Girl Club initiative which was implemented by the One Community project in Zomba and Machinga districts, two southern rural districts in Malawi. The Go Girl Clubs are GOCs that included VYA girls in health interventions such as SRH-related skills-based training including HIV prevention, education support, social skills, asset building, and economic strengthening. Starting in the year 2015 the Go Girl clubs were implemented under the **D**etermined, **R**esilient, **E**mpowered **A**IDS-free, **M**entored, and **S**afe lives (DREAMS) initiative which was funded by the US Government President's Emergency Plan for AIDS Relief (PEPFAR), and aimed to significantly reduce new HIV infections among AGYW by 40% in 15 countries, including Malawi [32]. Any girl or young woman between the age of 10 and 24 was eligible to join the GOCs. The participants included orphans and vulnerable children, out-of-school girls, in-school girls, pregnant girls, married girls, and young mothers. The clubs were organized according to age groups: the 10-14-year old clubs; 15-19-year-old-clubs and 20-24year-old clubs. In the clubs, young women called mentor mothers used a DREAMS tool kit which is a form of curriculum to train the participants (AGYW aged 10–14) in various skills relating to their SH. The skills included building social assets, learning about their body, communication skills, their future and vision, contraceptives and services, STIs and HIV, and HIV testing [32]. The intervention also included economic strengthening programs for VYAs' parents or caregivers and parent/caregiver programs that increased their knowledge, skills, and confidence to talk to their children about sexual health and to monitor their children's activities [32]. In the clubs, the participants met once every week for a period of one year between July 2017 to July 2018. Evidence from the implementation science research measuring the impact of the initiative among adolescent girls and young women (AGYW) aged between 15 and 24 showed significant improvements in comprehensive knowledge about HIV, condoms, and significantly higher levels of support for equitable gender norms and higher relationship power among AGYW among other improvements [33]. However, the implementation science did not explore the outcomes of the initiative among the VYA girls aged 10–14.

## Theoretical framework

In this study we drew on Bronfenbrenner's Social Ecological Model (SEM) to explore SS and its influence on VYAs' knowledge of sexual relationships [34–37]. The model considers a person's (for e.g., a very young adolescent girl's) growth in the context of the relationships that make up her environment [35]. The first level is the individual level and includes factors such as a person's knowledge, attitude, perception, and self-efficacy. We assumed that these individual-level factors may have an influence on where a VYA girl sought SS and influence her ability to access SS. The second level is the interpersonal level and includes relationships such as family, peers, and other social networks. We assumed that these relationships could be sources of SS for VYA girls. The third level is the organizational or institutional and includes structures such as the school, youth clubs, and health facilities among others. We reasoned that such institutions could also be sources of SS. Furthermore, we thought that VYA girls' access to these institutions could influence their individual-level factors such as knowledge of where to

get SS and their perceptions towards various forms of SS. The exosystem includes characteristics of the society at large such as, healthcare policies, media, and the gender order. The firth level is the systems and structures level which comprises cultural values, customs, and laws [34–38]. We assumed that the systems and structure level factors may affect the other levels and in turn the quality of SS available to VYA girls.

The GOCs offered an intervention to reduce HIV risk and increase the agency of AGYW at multiple levels of the SEM [32, 39] For instance, at the individual level, GOCs facilitated the building of self-esteem, and provision of SH knowledge [39]. At the interpersonal level, the GOCs facilitated the building of social networks of VYA girls with peers, mentors, and adults [39]. At the institutional level the GOCs linked VYA girls to HIV services. Moreover, at this level, the GOCs provided a forum where girls were able to meet and discuss SH issues [39]. At the systems and structure level the GOCs provided information that dealt with gender norms and abuse [39].

In this study, we used qualitative in-depth interviews to explore whether there was any difference in SS received concerning sexual relationships between VYA girls who attended GOCs and those who did not. Specifically, we assessed whether attending the GOCs had any influence in terms of SS that VYA girls received on sexual relationships. Three research questions were explored: 1) what is the available SS concerning sexual relationships for VYA girls in rural Malawi? 2) what is the role and influence of SS on addressing sexual relationship issues? and 3) is there any difference in SS received between VYAs girls who attended GOCs and those who did not?

## Methods and materials

This was a descriptive comparative qualitative study that assessed the experiences of SS regarding sexual relationships among VYA girls aged 10–14 who did and did not participate in GOCs [40, 41].

### Study setting

This study was conducted in two rural southern districts of Zomba and Machinga in Malawi. In both districts, one in three inhabitants are youth between the ages of 10 and 24 [42]. The districts were selected due to high HIV prevalence (13% and 16.3% among 15–years 49-year-olds, in Zomba and Machinga respectively), HIV treatment gaps, high proportions of orphans and vulnerable children, high prevalence of early sexual initiation, high rates of childbearing during the teen years, and high school drop-out rates among girls [33].

In Zomba, two sites were selected for the study. The first was where the DREAMS Initiative was being implemented and consisted of a catchment area of 63 villages, where 88 GOCs operated. Only two out of 88 GOCs included VYAs with a total of 20 VYA girls. The second site was selected because DREAMS activities were not present and was approximately 60 kilometers away from the first site. Moreover, this site was selected because there was no existing HIV programming for VYAs at the time we conducted this study.

Two sites were also selected in Machinga. The first site was where the DREAMs project activities were taking place and consisted of a catchment area of 45 villages, where 75 GOCs operated. There was only one GOC with VYA girls (n = 10). The second site was purposively selected because DREAMS activities were not present and was approximately 50 kilometers away from the first site. Also, we chose to conduct the study in this site as there was no existing HIV programming for VYAs at the time we implemented this study.

## Sampling and data collection

Participants were recruited using purposive sampling [43]. There were two main categories of study participants. The first category was the DREAMS project's participants' who were aged 10–14 years. At the time of the interviews, participants had been club participants for about 12 months. Through the help of a project person working on the DREAMS project, prospective participants were identified and invited to participate in the study. The inclusion criteria were availability and willingness to participate, and the ability to communicate experiences and opinions in an articulate, expressive, and reflective manner. To assess their ability to communicate and express themselves, the VYAs were asked some informal ice-breaking questions (e.g., such as what they to do in their free time and why) before the informed consent procedures,. A total of 30 VYA girls aged 10 to 14 years were approached to participate in the interviews; however, only 23 VYA girls between the ages of 12 and 14 participated in the in the interviews. Seven VYAs club members in Machinga were interviewed; three club members declined to participate after indicating that they were uncomfortable doing the interviews. In each of the two clubs in Zomba, eight interviews were conducted with VYAs. Two club members from each of the two clubs were not available on the day the interviews were conducted.

The second category of respondents included VYA girls selected in communities that were not in the DREAMS implementation area. The District Youth Officer identified youth volunteers working in the communities, who assisted in identifying VYAs who were eligible for the study. VYA girls were approached and invited to participate in the study. Our inclusion criteria included VYA girls aged 10–14, who had not participated in any youth club before. For each site, an initial number of 10 respondents was decided to match the numbers in the DREAMS sites with an allowance of increasing the sample in case saturation could not be reached. However,10 respondents at each site were sufficient to reach saturation and a total of 20 VYA girls were interviewed.

The interviews were done in October 2019 by the first author who is a female researcher with experience working with adolescents. Interviews were conducted in Chichewa, a dominant spoken language in the study settings. The interviews were conducted face-to-face in the community where the respondents lived before COVID-19. To ensure privacy and confidentially, a private house was arranged within the community where individual interviews took place. No one else was present during the interview apart from the interviewer and the respondent. A semi-structured interview guide for IDIs was framed around questions that explored the role of SS in sexual relationships. The first part of the interviews focused on building rapport while also gathering some information about the respondents. These included questions about time use, mobility, SS, and networks. The second part of the interview focused on asking respondents to share stories about sexual relationship experiences. These questions asked whether sexual relationships among VYA happen in their communities, why they happen, whether the respondent has ever been in a sexual relationship, why they were in sexual relationships, their opinions of sexual relationships, and experiences of the support they received from friends, parents, relatives, and others if they have questions regarding sexual relationships. Interviews occurred one time, lasted between 30–40 minutes, and were recorded using a digital recorder.

## Data analysis

In preparation for data analysis, audio-recorded IDIs were transcribed verbatim. Transcripts were then carefully translated from Chichewa into English. The first author listened to Chichewa recordings and checked against the translated transcripts to make sure that the transcripts did not lose their original meanings. The transcription and translation were done by a

hired experienced research assistant. To check for quality, the first author read the transcripts and checked the transcripts against the audios. The transcripts were then imported into the NVIVO software (NVIVO12, QSR International).

A constant comparative analysis (CCA) approach was used to analyze the data. This analytic method entails a set of systematic procedures relating to assigning codes or categorizing data and subsequently identifying themes or patterns [40, 41]. The rationale behind using the CCA is that it provides a logical approach to examining and understanding qualitative data where there is a need to do a comparison, for example, comparing a group of respondents, codes or themes [44]. In addition, CCA guarantees that every piece of data is systematically compared to every other piece of data in the data set [45]. In this study, the CCA allowed us to systematically compare data from the GOC participants with data from the Non-GOC participants. The first author repeatedly read through all transcripts to familiarize herself with the content and to develop a list of themes. On subsequent readings of the five transcripts, the first codes reflective of key concepts in the transcripts were developed. Relationships between the codes were established, and codes with common elements were combined into categories. A preliminary codebook was developed where codes and code definitions were made guided by both the data and the research questions whereby we looked for patterns and themes about the concept of SS in sexual relationships. To explore the concept of SS categorized SS into four categories informational, instrumental, emotional and transactional SS as described in Barker's review of 2007 [29]. Codes were grouped into categories, then developed into themes, and analyzed using the Nvivo 12 software. An independent researcher was engaged in coding and this enabled us to assess if there was an inter-coder agreement [43, 46]. A coding comparison query was run in Nvivo 12 using a percentage agreement test and an 80% agreement was achieved.

All data were first analyzed separately (i.e., VYA girls in GOCs and VYA girls not in Clubs) before comparing and contrasting themes within and between the two different groups of participants. The codes and categories were then refined to represent the data more accurately. Quotes illustrating meaning or key message from the analysis were selected and used with narratives to explain the data based on how best they represented the key message.

### Ethical considerations

Ethical approval to conduct this study was obtained from the College of Medicine Research Ethics Committee (COMREC) in Malawi (Ethics Approval Number: PP.01/17/2095), and from the University of the Witwatersrand Human Research Committee (HREC) in South Africa (Approval number: M181009) and from Population Council in the United States of America (Approval number: 784). Guidelines related to interviewing children and adolescents were followed to ensure that the study was conducted ethically and that the procedures followed by the study protected the participants and minimized harm [47]. Before engaging the VYA girls a parent or guardian was approached to obtain her/his permission to interview the girl using an appropriate informed consent document. After the parent/guardian consent to the VYA girl's participation, the VYA girl was approached to provide assent.

## Results

### Social demographic characteristics of study participants

A total of 43 VYA girls were interviewed (Table 1). The age of the respondents ranged between 10 and 14. The youngest of the respondents, ages 10–11, were not GOC participants. The majority of the respondents had never been in a sexual relationship (70% GOC participants, 90% not GOC participants) and were in primary school (78% GOC participants, 80% not

**Table 1. Social demographic characteristics of participants.**

| | GOCs | | Not in GOCs | |
|---|---|---|---|---|
| Characteristic | Number (N = 23) | % | Number (N = 20) | % |
| **Age** | | | | |
| 10 | 0 | 0 | 1 | 5 |
| 11 | 0 | 0 | 2 | 10 |
| 12 | 5 | 22 | 4 | 20 |
| 13 | 9 | 39 | 5 | 25 |
| 14 | 9 | 39 | 8 | 40 |
| **Education Status** | | | | |
| In school (Primary school) | 18 | 78 | 16 | 80 |
| Out of school | 5 | 22 | 4 | 20 |
| **Housing (living with)** | | | | |
| Both Parents | 15 | 65 | 6 | 30 |
| Single Parent | 5 | 22 | 4 | 20 |
| Other relatives | 3 | 13 | 10 | 50 |
| **Sexual relationships** | | | | |
| Ever been | 7 | 30 | 2 | 10 |
| Never been in any | 16 | 70 | 18 | 90 |

GOC participants). The majority of club participants were living with both parents while most of the non-club participants were living with relatives.

## Available social support for VYA girls

Using Barker's definition of SS, we explored what types of SS concerning sexual relationships were available for VYA girls in our study site. Informational SS was the prevalent type of SS among VYA. Barker defines informational SS as an information that which an individual obtains from other people about where they can get help e.g. a health service, to protect themselves from negative health outcomes [29]. Among both the club participants and non-participants, the respondents reported they received support in the form of information or advice about sexual relationships from various sources including parents, siblings, relatives, and friends. Respondents who were in GOCs also reported informational support from the GOCs they attended. Three main themes concerning informational SS came out: information support from parents/older relative, information support from clubs, and information support from friends.

## Information support from parent/older relatives

Respondents were asked whether they had ever had any conversation about sexual relationships with anyone or sought information about sexual relationships. Among club participants and non-participants, respondents indicated they received some information about sexual relationships from parents and older relatives. There were no major differences between club participants and non-participants in terms of information support received from parents and older relatives. Respondents in both settings indicated that parents and older relatives mostly disapprove of sexual relationships among the VYA girls and that information offered by these sources is mostly advice on the risks of sexual relationships such as contracting HIV and unwanted pregnancies to keep VYA girls away from boys.

*"As for me; I don't take part in them [sexual relationships] totally because my parents advise me and I take heed of the advice (of not engaging in sexual relationships), because if I wasn't taking heed then I would have been pregnant or contracted diseases. Because the friends that I had, I was playing with them and we are of the same age but as for now some of them have one child each some are pregnant."*

[**Non-club Participant**, **Age 12, Zomba**]

*"On relationship issues, grandma just says "don't do what? Don't have sexual intercourse with boys. In this world, there are sexually transmitted diseases. If you want to be doing that, maybe you should be using condoms." And also her, she does not want this kind of behavior.***"**

[**Club Participant**, **Age 13, Machinga**]

Differences were noted however between club participants and non-club participants when they were asked if they ever consult parents or older relatives when they have questions about sexual relationships. Among non-club participants, several respondents reported being uncomfortable discussing with parents about sexual relationships because of fear and disapproval from parents.

Interviewer: What about on the part of sexual intercourse? With your mother have you ever talked about this issue before?

Respondent: No. she has never told me.

Interviewer: She has never told you?

Respondent: Yes.

Interviewer: What about your father?

Respondent: Umm, my father I fear him.

[**Non-club Participant**, **Age 12, Machinga**]

Among the GOC participants, however, most respondents reported having consulted an older relative for decision-making regarding sexual relationship issues. For instance, some respondents reported consulting older relatives to seek help in decision-making on whether to have sex with their boyfriend. In the quote below a respondent narrated an incident where she enlisted the help of a relative when her boyfriend was demanding sex:

*"I am sometimes able to ask my sister-in-law (wife to her brother) because this other time I was in a relationship with a certain boy, and he loved sexual intercourse, right? So I said that me; I said that "I don't want," and he said that "if you refuse, me and you, our relationship will end." I agreed that even if it ends. I came here I asked; I asked my in-law. She said "Aah you; leave him, if he is talking about that, leave him. There is something that he knows, don't waste time with him." So I stopped it [relationship].***"**

[**Club Participant**, **Age 14, Machinga**]

## Information support from GOCs

Apart from mentioning parents and older relatives as sources of information on sexual relationships, respondents in GOCs indicated that they also received informational SS from clubs.

Getting additional informational support from GOCs was the main distinguishing factor between the two groups of participants. When compared to non-GOC participants, club participants reported learning about safe sex, pregnancy and how to prevent it, and how to reduce the risk of acquiring HIV and other sexually transmitted infections.

*"..at the clubs, we learnt about contraception, that we should use condoms during sex, and prevention of sexually transmitted diseases and prevention of pregnancy."*

[**Club Participant**, **Age 12, Zomba**]

*"I have benefited [from attending the DREAMS GOCs], right? Because I did not know that us; pregnancy how can we avoid it? Or AIDS how can we avoid it? We did not know. . ...So we learned it there at One Community (DREAMS GOCs)."*

[**Club Participant**, **Age 13, Machinga**]

Another difference and advantage that the GOC participants had over non-club participants were that they reported having opportunities to consult or ask sensitive questions about sexual relationships to club facilitators. GOC participants reported being comfortable asking these questions because the facilitators were people who taught them about SRH issues.

Interviewer: Okay. I want to know questions like what do you ask your facilitator? Just for example.

Participant: For example, like maybe a person is asking me to be his girlfriend and I am refusing him, right?

Interviewer: Yes,

Participant: . . .so there was this boy who has been disturbing me. To listen to my refusal, he wasn't listening but continued blocking me on the way. So I told the facilitator saying "oh that boy, for me to walk; how should I walk? because when I go this way, this person is blocking my way. How should I walk around?" So she told me that if he touches you; If he touches you, you should scream. And then I did that and people came, they advised him not to trouble me gain, that's how it ended."

[**Club Participant**, **Age 12, Machinga**]

## Information support from friends

Another source of information support reported by respondents was friends. This was reported by the club and non-club participants. In both study settings, several respondents reported having had a discussion about sexual relationships with friends. Unlike the information support received from parents, older relatives, and clubs, information sourced from friends seemed to carry conflicting messages. For instance, while some respondents indicated friends advised them that sexual relationships are bad, other respondents indicated friends encouraged them to engage in sexual relationships.

*"I discuss with my friends the disadvantage and advantages of being in a relationship. The disadvantage of having a relationship is that one may be forced by her partner to have sex and maybe she also wants, one ends up destroying her future by getting pregnant. In other relationships other partners don't think about that, they help each other with school only."*

[**Non-club Participant**, **Age 13, Machinga**]

"*My friends were telling me that in sexual relationships they are given money and enough love. . ..and also there are times you meet a boy and is proposing to you and you said no, and your friends might be saying why did you say no, and they will tell you that you didn't think properly to say no.*"

[**Non-club Participant**, **Age 13, Zomba**]

In both study settings several respondents who were in a sexual relationship reported that they had been influenced by their friends to engage in these relationships. The influence from friends was indirect and occurred in two ways. The first was they admired the benefits their friends received from being in a relationship such as receiving money, soap, or body lotions from boyfriends. The second was they wanted to be like other girls who were in relationships.

"*I was worried that I sometimes went go to school without eating, I could just take a bath and end up not applying lotion and went to school so my friends were laughing at me that I should find a boyfriend that will be giving me money to buy every material needed in my life so they didn't allow me to be their friend.*"

[**Non-club Participant**, **Age 13, Machinga**]

Even though the VYA girls from GOCs reported peer pressure and the existence of conflicting messages from friends, the information they had received from the clubs helped most of them navigate through sexual relationships issues including misinformation. For instance, some GOCs participants reported that despite being encouraged by friends to engage in sexual relationships, they followed the advice received at the GOCs which discouraged them from engaging in sexual relationships.

"*. . ..when I have gone to chat with my friends, my friends tell me that I should go and engage in a sexual relation. . .I should go and engage in a relationship but when I come here[at the clubs] they discourage me from doing so; what I do is that I follow the advice that I am provided with here. When I hear about HIV, I am afraid of HIV because I heard at the clubs that HIV is dangerous and it can kill people so I am now afraid of this.*"

[**Club Participant**, **Age 13, Machinga**]

Furthermore, most of the GOC participants reported discussing with friends what they learned in clubs on sexual relationships. Several club participants reported discussing information such as condom use, STI and HIV prevention, and pregnancy prevention.

"*Maybe when you want to have sexual intercourse with a boy, you should, first of all, see the [condom's] expiry date. That this condom is it still strong? Has it not expired? Then when you see that the condom is okay, you can tear it and see the inside that is there air? Then when you see that the inside is okay, then whether your boyfriend or husband if you feel comfortable with him, you can dress him. Yes; that's what we discuss with my friends.*"

[**Club Participant, Age 13, Machinga**]

"*We [with friends] talk about an issue that we discuss at Go-girls[GOCs]. . .We learned about 'my dreams' and we talk to each other that it is one's idea to choose any job that one wishes to*

*do when one finish education. And we discuss visions, a girl who is focused. And we also learned how to prevent HIV, use of condoms and use of contraceptives.*"

[**Club Participant**, **Age 12, Zomba**]

## Discussion

This study aimed to explore if there was any difference in SS received concerning sexual relationships between the VYA girls who attended GOCs and those who did not. Specifically, we wanted to see if attending the GOCs had any influence in terms of SS that VYA girls receive on sexual relationships. In both study settings, the main type of SS received regarding sexual relationships was informational support. However, the GOC participants and Non-GOC participants however differed in the way they utilized the informational SS to navigate through sexual relationship issues. The GOCs were an additional source of informational SS reported by the GOC participants.

According to the SEM we found that the mesosystem level which includes parents, older relatives, friends and GOCs were the main sources of informational SS regarding sexual relationships. This finding is pointing to the need for VYA SH programs to include interventions that target this level. The finding that parents, older relatives, and friends were the main sources of SH information is consistent with literature which shows that parents, relatives, and friends are the main sources of sexual health information for adolescents [2, 11, 15, 48, 49]. Our findings indicate that in both study settings information provided by parents and older relatives was often in the form of advice against engaging in sexual relationships. However, other respondents especially the non-club participants reported receiving no advice or having any discussion with parents or caregivers. While advice against sexual relationships may be a positive thing in terms of keeping VYA girls from negative SRH outcomes such as pregnancies and sexually transmitted diseases, fear of parents' reactions such as anger or dismissal may prevent some VYAs from opening up with parents or older relatives about sexual relationships. For instance, we found out that some respondents especially among the non-club participants reported having had no talk or discussion about sexual relationships with parents because of fear of disapproval. On the other hand, club participants reported having had consulted parents or older relations for decision-making regarding sexual relationships. This finding suggests the influence of club attendance in empowering the VYA girls who attended clubs with confidence to seek out information. In addition, the club participants also reported consulting club facilitators for decision-making regarding sexual relationships suggesting another important role of the clubs in providing an alternative intervention that can facilitate open and free communication for VYAs, especially on sexual health issues. Similar findings have been reported in other studies where the DREAMS interventions have been implemented. For example, our findings reflect those of studies among foreign migrant AGYW aged 14–19 in South Africa and among the AGYW aged 13–22 in Kenya [50, 51]. These findings suggest the need for alternative sources of information for VYAs.

Our findings indicate that among both the club participants and non-participants, friends were an important source of information about sexual relationships. However, VYA girls reported getting conflicting messages about sexual relationships from friends whereby some friends encouraged them to engage in sexual relationships and some discouraged them. This finding suggests a possibility of VYAs being misled or misinformed by friends on sexual relationship issues. This is in agreement with literature that documents that in early adolescence misinformation about sexuality issues abounds from peers, siblings, and other unreliable sources of information [2, 11, 52, 53]. The findings suggest the need for intervention

specifically for VYA girls with a focus on providing correct information on sexual relationship issues. In this study, we observed that club participants were able to use information accessed from the clubs to correct misinformation about sexual relationships they got from friends. In addition, club participants reported having received more information about sexual relationship issues from the clubs and reported use of this information in sexual relationships, such as condom use. This is a very important finding considering that in rural settings, few places provide accurate and comprehensive sex education to VYAs. Furthermore, the findings are also relevant when we consider that out-of-school VYAs usually have no access to sex education [54–56]. Our study included out-of-school VYA girls and shows that interventions like GOCs can provide access to such. Therefore, increasing avenues such as GOCs where VYA can access correct information about sexual health issues is critical for developing healthy sexual well-being of VYAs in rural settings.

The concept of GOC is not entirely a new approach as it has been implemented before [57–59]. However, this study brings in findings from an implementation science study with a unique focus on VYA girls. Until recently, most adolescent SRH programs tended to focus on older adolescents (15–19 years). This study adds to knowledge on a few VYA programs that have been implemented. Our findings are especially critical for VYA girls when the 'SRH and gender' dividend is taken into consideration whereby intervening at this age to strengthen their social and protective assets, will enable them to protect themselves from negative sexual health outcomes including reducing their risk for HIV.

Our findings have important implications on adolescent SRH programming in Malawi, considering that studies have highlighted challenges on the delivery of SH information to VYAs in SSA countries including Malawi [60–63]. Some of the challenges include prohibitive SH messages from parents and other adults, some teachers in schools not adequately equipped to deliver SH information to VYA, some teachers being selective on what SRH topics they teach to VYAs and their emphasis on abstinence [60–63]. Our findings suggest that VYA SRH programming that include alternative sources of information such as the GOCs could be beneficial to VYAs. Highlighting the important role played by the GOCs, our findings suggest that several VYA girls who were in sexual relationships obtained accurate knowledge from the GOCs which they used to make decisions in their sexual relation relationships. Secondly SRH programming for VYAs should include social asset building which includes creating of an enabling environment for VYA through training of near-to-peer models such as club facilitators in GOCs to allow VYAs to seek out information and learn SH issues comfortably. Moreover, social asset-building could include empowering parents and other adults on how to talk about SH issues with VYAs. The asset-building framework posits that adolescent girls need a combination of social, health, cognitive, and economic assets to make a safe and healthy transition from childhood to adulthood [50, 64, 65]. Social asset building intervention have been found to increase agency in VYAs thereby reducing their vulnerability to HIV [50, 64, 65]. Our findings have SRH programming implications for VYAs who are not yet sexually active. We have found that younger adolescents seem to benefit from having access to SRH programmes at their age.

The strength of this study lies in the inclusion of a comparison of non-beneficiaries of the DREAMS intervention. This inclusion allowed us to compare and contrast the influence of the GOCs as SS the VYA girls received on sexual relationship issues.

The study has some limitations. First, only one interview was completed for each respondent. The amount of time for each interview was between 30 to 45 minutes. The total amount of time spent with each respondent was affected by the age of the respondents. Being young adolescents, most of the respondents did not provide a lot of details in their responses. In addition, the age difference between the researcher and the young adolescents participating in the

study could also have limited how much the respondents said. All these factors may have limited the study since it reduced the time for the respondent to say more and the time for the interviewer to probe deeper. A second limitation is that we did not explore other stakeholders' perspectives, such as club facilitators, parents, and other family relations to the club participants. A third limitation is that there is a possibility of a social desirability bias considering the nature of the research topic on sexual relationship issues. A fourth limitation is that while we conducted a comparative analysis between the GOC and Non-GOC participants with regards to the SS they received regarding sexual relationships other factors (e.g., difference in primary caregivers) could have made for a difference in answers provided by the VYAs interviewed in this study.

## Conclusion

This study compared the GOC participants and non-club participants in terms of the SS they received regarding sexual relationships. Parents, other relatives, friends, and GOCs were found to be the main sources of SS and this SS was provided inform of information. We found that the GOC participants had received additional SS from the clubs they attended which made them more knowledgeable and to be better at handling sexual relationship issues than their counterparts who were not in clubs. Our study findings suggest that increasing avenues such as GOCs where VYA can access correct information about sexual health issues is critical for protecting VYA from negative SRH outcomes including HIV. Furthermore, interventions that integrate SS including social asset building and safe spaces should be considered in SRH programming for VYAs.

## Supporting information

**S1 Appendix. In-depth interview guide for very young adolescent girls (10–14) in clubs.**
(DOCX)

**S2 Appendix. In-depth interview guide for very young adolescent girls (10–14) not in clubs.**
(DOCX)

**S1 Data. Anonymized data.**
(ZIP)

## Acknowledgments

The authors would like to thank the DREAMS Implementation Science research team, including colleagues at the Population Council and Centre for Reproductive Health at the University of Malawi, College of Medicine, who made this work possible. Special thanks to One Community for facilitating access to their program participants and for sharing their time and expertise. Thanks to all the girls who participated in the study who committed their time and shared their experiences and opinions.

## Author Contributions

**Conceptualization:** Wanangwa Chimwaza-Manda, Mphatso Kamndaya, Nanlesta Pilgrim, Yandisa Sikweyiya.

**Formal analysis:** Wanangwa Chimwaza-Manda, Nanlesta Pilgrim, Sanyukta Mathur, Effie Kondwani Chipeta, Yandisa Sikweyiya.

**Methodology:** Wanangwa Chimwaza-Manda, Mphatso Kamndaya, Sanyukta Mathur, Effie Kondwani Chipeta, Yandisa Sikweyiya.

**Software:** Wanangwa Chimwaza-Manda.

**Supervision:** Mphatso Kamndaya, Nanlesta Pilgrim, Yandisa Sikweyiya.

**Writing – original draft:** Wanangwa Chimwaza-Manda.

**Writing – review & editing:** Wanangwa Chimwaza-Manda, Mphatso Kamndaya, Nanlesta Pilgrim, Sanyukta Mathur, Effie Kondwani Chipeta, Yandisa Sikweyiya.

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
