## [Decision Letter · Decision Letter 0]

10 Aug 2022

PGPH-D-22-00488

Social support and its influence on sexual relationships among very young adolescent girls: A comparative qualitative study of Girl Only Clubs’ participants and non-participants in rural Malawi

Dear Dr. Manda,

Thank you for submitting your manuscript to PLOS Global Public Health. After careful consideration, we feel that it has merit but does not fully meet PLOS Global Public Health’s publication criteria as it currently stands. Therefore, we invite you to submit a revised version of the manuscript that addresses the points raised during the review process.

We look forward to receiving your revised manuscript.

Kind regards,

Tia M. Palermo

Academic Editor

Journal Requirements:

1. Please update your online Competing Interests statement. If you have no competing interests to declare, please state: “The authors have declared that no competing interests exist.”

2. Please amend your detailed online Financial Disclosure statement. This is published with the article. It must therefore be completed in full sentences and contain the exact wording you wish to be published.

Please state what role the funders took in the study. If the funders had no role in your study, please state: “The funders had no role in study design, data collection and analysis, decision to publish, or preparation of the manuscript.”

3. In the Funding Information you indicated that no funding was received. Please revise the Funding Information field to reflect funding received.

Please ensure that the funders and grant numbers match between the Financial Disclosure field and the Funding Information tab in your submission form. Note that the funders must be provided in the same order in both places as well.

4. "Appendix 1a English IDI Guide_Girlsinclubs.docx" and "Appendix 1b English IDI Guide_Girlsnotinclubs.docx" are currently uploaded as file type “Other”, which is not viewable by the reviewers. Please change the file types to 'Supporting Information' and include a legend in the manuscript if you wish them to be included in review. 

5. In the online submission form, you indicated that “The interview data of this study are available from the corresponding author upon reasonable request.”. All PLOS journals now require all data underlying the findings described in their manuscript to be freely available to other researchers, either 1. In a public repository, 2. Within the manuscript itself, or 3. Uploaded as supplementary information.

Additional Editor Comments (if provided):

Reviewers' comments:

Reviewer's Responses to Questions

**Comments to the Author**

1. Does this manuscript meet PLOS Global Public Health’s publication criteria? Is the manuscript technically sound, and do the data support the conclusions? The manuscript must describe methodologically and ethically rigorous research with conclusions that are appropriately drawn based on the data presented.

Reviewer #1: Partly

Reviewer #2: Yes

2. Has the statistical analysis been performed appropriately and rigorously?

Reviewer #1: Yes

Reviewer #2: N/A

3. Have the authors made all data underlying the findings in their manuscript fully available (please refer to the Data Availability Statement at the start of the manuscript PDF file)?

Reviewer #1: No

Reviewer #2: Yes

4. Is the manuscript presented in an intelligible fashion and written in standard English?

Reviewer #1: Yes

Reviewer #2: Yes

5. Review Comments to the Author

Reviewer #1: Manuscript Review: Social support and its influence on sexual relationships among very young adolescent girls: A comparative qualitative study of Girl Only Clubs’ participants and nonparticipants in rural Malawi

Study assesses the difference in social support (SS) received concerning sexual relationships between the very young adolescent girls who attended Girl Only Clubs and those who did not using a qualitative approach of thematic and descriptive analysis.

The authors assess an important and until now relatively underanalyzed topic: the influence of Girl Only Clubs on social support around sexual relationships among very young adolescent girls. They use a comparative analysis which has advantages in understanding how social support differ between girls who participated in the clubs and those who had not. The authors find that there is a potential important role for Girls Only Clubs or similar interventions to address the current information gap for very young adolescents to get reliable information and a safe space to discuss sexual relationships. I enjoyed reading this study, and I think there is need in the current literature to add to this topic. However, there are three key points in which the manuscript should be improved. First, the writing could be strengthened throughout the manuscript with clearer definitions of key concepts earlier in the text and an overall proofreading of sentence structure and typos. Second, the authors should add a section on the reasons why they selected their current methodology for a comparative analysis and what the added value is for this study. Third, the current study could use a stronger theoretical framework linking existing literature on very young adolescents and sexual and reproductive health outcomes, the role of girls’ clubs (or similar interventions) and social support, and the connection between social support and health outcomes.

Below are some further considerations by section.

Writing and organization

• I highly suggest having a person not involved with the study to proofread the manuscript for clarity. The authors 1) use acronyms such as SHR, AGYW before explaining them in full; 2) introduce essential concepts such as ‘very young adolescents’, ‘Girls Only Clubs’ without defining them at first use.

• The title is kind of misleading. The study is mostly on the interaction between Girls only clubs and social support and the knowledge young adolescents have about sexual relationships, but not about the actual sexual relationships themselves. Consider making this more specific.

• I would avoid the opening sentence as it is currently used in the abstract. The claim that ‘there is growing evidence that some are’ is not specific and could be said more convincingly by focusing for instance on changing trends in age of sexual debut or the possible consequences of early sexual relationships on health and wellbeing outcomes.

Introduction

• Normally the introduction would also include a preview of the methods, findings and possibly a guide to introduce the reader to what to expect in the rest of the study.

• The underlying theoretical framework is lacking. While I understand that there is limited research available on very young adolescents, social support and sexual relationships, the authors should at least set out the relationship between components that have been covered in the existing literature more clearly, in specific 1) the connection between social support and sexual relationships; 2) very young adolescents and sexual and reproductive health outcomes, 3) girls clubs or similar interventions and social support. I recommend using a logical framework or a theory of change to clearly highlight the pathways in which they expect the Girls Only Clubs and social support to have an influence. Using a framework will also help to clarify how the authors view the interaction between social support and the Girls Only Clubs.

• The authors could further strengthen the literature used in the study. While Barker’s literature review is salient to this manuscript, he did not develop the categories on social support. Further review of the original literature, as well as including more diversity in perspectives is needed to strengthen the background section.

Intervention

• I would like to see more information on the intervention: the Girls only Clubs and DREAMS initiative. Information such as how girls were selected for the program, how the information was transferred to the adolescents (e.g. who was leading the clubs, how often would they meet, what were the other benefits from attending).

Methodology

• The justification and explanation for the choice of methodology is insufficient and should be addressed. The authors do not give any reasons for why they chose to assess their research questions using a comparative analysis. They should also add further explanation what the benefits of this methodology is for the study.

Study setting

• Additional information on the non-Girl Only Club locations and sample is required. Were there other considerations other than the lack of the intervention (e.g. the sampling section suggest that no other HIV programming was present, if this was a consideration for selection it should have been included in ‘study setting’ section.

Sampling

• Describe how the inclusion criteria were assessed, in particular with regards to ‘the ability to communicate experiences and opinions in an articulate, expressive, and reflective manner’. Also, currently it seems that all 7 girls who were not interviewed were below 12 years. Rephrase this section if this was not the case.

Data analysis

• Add who were involved in coding and what the outcomes of the test for inter-coding agreement was (and if applicable what the adjustments and consequences are).

Results

• Even though this is a comparative analysis, the sample sizes are small and this is a descriptive analysis only. For that reason be cautious about using general statements about Girls Only Club girls being more confident in their knowledge and having more to say (p.13). I recommend focusing more on the type of information, social support received rather than reporting which group received more. If you want to use the frequency, it would be good to be more specific. Use something like ‘x out of 23 Girls Only Club girls expressed being confident about their knowledge, while this was only so for x out of 20 non-Girls Only Club girls’.

Discussion

• The other do a good job describing the findings in the discussion and putting them in context. When using additional literature, be careful to clarify why the literature was different (see e.g. line 385 add if the findings from the other DREAMS study are only for older adolescents).

• With regards to the two main implications, the first part of the first implication does not follow directly from the analysis. You are not assessing whether adolescents should get SHR programs at a younger age. You can say that you found that younger adolescents seem to benefit from having access to these programs at their age. It is a subtle difference, but since you are not doing a comparative analysis with older adolescents I find the first part of the implication a bit of a stretch. I agree with the second implication, but please reread it to correct the language.

• On the limitations, given that the study claims its main strength is the comparative analysis, the authors should also acknowledge any limitations in that respect. E.g. difference in primary caregivers could make for a difference in answers.

Conclusions

• In line 442 and 443 when you mention that there is ‘no difference in SS’, it would be good to be more specific, because currently it contradicts with the rest of the sentence.

Reviewer #2: Several typos were noted, and some awkward phrasing (especially the paragraph beginning on line 415) made it difficult to read in parts. A thorough proofread would be beneficial.

Overall, the findings were interesting and analysis appears to be thorough and rigorous. From line 168: a bit more detail in terms of coding and analysis would be helpful - why was the specific methodology (constant comparative analysis) chosen? How was it ensure that translations did not lose their original meanings?

Line 215 - the definition of informational SS is a bit vague. Since it's so critical to the findings, expanding on this concept might be helpful to readers.

You introduce the concept of social asset building at the end of the paper (around line 415) without explaining or elaborating. Since you also include this as a recommendation for future programming, it would be helpful to explain what it is and why it is so important to programming.

The findings are very rich, and could help to inform future programming. As such, it would be worth considering adding a bit more information on the importance and the challenges of providing SRH education to VYA - e.g., protective factors of confidential adults (club facilitators, parents, etc.) are mentioned, but consider expanding on why these are important. More discussion about the challenges of delivering SRH to VYA who are not yet sexually active (which the majority of your sample appeared not to be), or how this information might have affected this study would also help to bolster recommendations for future studies/programming.

6. PLOS authors have the option to publish the peer review history of their article (what does this mean?). If published, this will include your full peer review and any attached files.

**Do you want your identity to be public for this peer review?** For information about this choice, including consent withdrawal, please see our Privacy Policy.

Reviewer #1: **Yes: **Marlous de Milliano

Reviewer #2: No

---

## [Editor Report · Decision Letter 1]

8 Nov 2022

PGPH-D-22-00488R1

Social support and very young adolescent girl’s knowledge on sexual relationships: A comparative qualitative study of Girl Only Clubs’ participants and non-participants in rural Malawi

Dear Dr. Chimwaza Manda

Thank you for submitting your manuscript to PLOS Global Public Health. Thank you for the careful revisions of your manuscript. I am writing to conditionally accept this manuscript, provided you make the following changes first:

Abstract

Change: “Early sexual relationship is” to “early sexual relationships are”

Remove extra space between “use” and “club”

Introduction

Move description of intervention in lines 99-110 from intro to methods section if this information is referring to the current study (and not DREAMS overall) under a sub-heading of programme description

We look forward to receiving your revised manuscript.

Kind regards,

Tia M. Palermo

Academic Editor

Journal Requirements:

Additional Editor Comments (if provided):

Abstract

-Change: “Early sexual relationship is” to “early sexual relationships are”

-Remove extra space between “use” and “club”

Introduction

-Move description of intervention in lines 99-110 from intro to methods section if this information is referring to the current study (and not DREAMS overall) under a sub-heading of programme description

Reviewers' comments:

N/a

---

## [Editor Report · Decision Letter 2]

28 Nov 2022

Social support and very young adolescent girl’s knowledge on sexual relationships: A comparative qualitative study of Girl Only Clubs’ participants and non-participants in rural Malawi

PGPH-D-22-00488R2

Dear Ms Manda,

I am pleased to inform you that your manuscript 'Social support and very young adolescent girl’s knowledge on sexual relationships: A comparative qualitative study of Girl Only Clubs’ participants and non-participants in rural Malawi' has been provisionally accepted for publication in PLOS Global Public Health.

Best regards,

Tia M. Palermo

Academic Editor
